# Improved Performance and Bias Stability of Al$_2$O$_3$/IZO Thin-Film Transistors with Vertical Diffusion

**Se-Hyeong Lee, So-Young Bak and Moonsuk Yi ***

Department of Electrical and Electronics Engineering, Pusan National University, Busan 46241, Korea; shlee19@pusan.ac.kr (S.-H.L.); bso6459027@pusan.ac.kr (S.-Y.B.)
* **Correspondence:** msyi@pusan.ac.kr; Tel.: +82-51-510-2381

**Abstract:** Several studies on amorphous oxide semiconductor thin-film transistors (TFTs) applicable to next-generation display devices have been conducted. To improve the poor switching characteristics and gate bias stability of co-sputtered aluminum–indium–zinc oxide (AIZO) TFTs, we fabricate Al$_2$O$_3$/indium–zinc oxide (IZO) dual-active-layer TFTs. By varying the Al$_2$O$_3$ target power and oxygen partial pressure in the chamber during Al$_2$O$_3$ back-channel deposition, we optimize the electrical characteristics and gate bias stability of the Al$_2$O$_3$/IZO TFTs. The Al$_2$O$_3$/IZO TFTs, which are fabricated under 50 W Al$_2$O$_3$ target power and 13% oxygen partial pressure conditions, exhibit a high electron mobility of 23.34 cm$^2$/V·s, a low threshold voltage of 0.96 V, an improved on–off current ratio of $6.8 \times 10^7$, and a subthreshold swing of 0.61 V/dec. Moreover, by increasing the oxygen partial pressure in the chamber, the positive and negative bias stress values improve to +0.32 V and −2.08 V, respectively. X-ray photoelectron spectroscopy is performed to reveal the cause of these improvements.

**Keywords:** amorphous oxide semiconductor; thin-film transistor; carrier suppressor; RF-magnetron sputtering; vertical diffusion

## 1. Introduction

Over the past few years, amorphous oxide semiconductors (AOS) such as amorphous indium–zinc oxide (a-IZO) have been studied extensively as channel layer materials for thin-film transistors (TFTs) because of their high electron mobility, transparency in visible light, uniformity of performance over a large area, and compatibility with flexible substrates. AOS TFTs are expected to be utilized in next-generation display devices, such as high-resolution, high-scanning-rate, flexible, and transparent devices [1–3]. High electron mobility is achieved because of the overlap of the s-orbitals of the metal, and electron carriers are supplied through the ionization of oxygen vacancies in the channel layer. However, a-IZO TFTs exhibit poor switching characteristics and voltage-bias stability because oxygen vacancies trap electron carriers. In amorphous indium–gallium–zinc oxide (a-IGZO), the most widely used AOS material in the channel layer of TFTs, the Ga cation improves the performance and bias stability by suppressing the generation of oxygen vacancies, which are traps in the channel layer and channel–insulator interface [4–6]. However, Ga is a rare earth element, and it is costly to be used for mass production. Several studies have been conducted to identify substitutes for expensive rare elements to resolve this cost limitation. Metal elements such as Ga, Al, Hf, and Zr have low electronegativity, standard electrode potential, and high metal–oxygen bonding energy [7–10]. These elements in the indium–zinc-based AOS channel layer, called carrier suppressors, form bonds with oxygen and suppress the generation of oxygen vacancies [11,12].

Because Al not only has a high metal–oxygen bonding energy but also has abundant reserves on Earth, we fabricated co-sputtered aluminum–indium–zinc oxide (AIZO) TFTs in a previous study. These devices exhibited good electrical performance and bias stability

but poor switching characteristics [13]. Because the radius of the Al cation is shorter than that of the In and Zn cations and equal to that of the Si cation, the Al cation easily migrates in the channel layer and serves as a trap defect at the channel–insulator interface [14–16]. In this work, to improve the switching characteristics and bias stability of AIZO TFTs, we fabricated AOS TFTs with an $Al_2O_3$/indium–zinc oxide (IZO) channel layer structure using vertical diffusion techniques [17,18]. By depositing an $Al_2O_3$ layer onto the IZO channel layer, the Al cations diffused into the IZO front-channel layer during the annealing process and suppressed the generation of oxygen vacancies. In addition, by varying the conditions of the $Al_2O_3$ back-channel deposition, such as the $Al_2O_3$ target power and oxygen partial pressure (OPP) in the chamber, we optimized the electrical characteristics and bias stability of the fabricated devices.

## 2. Materials and Methods

Inverted-staggered-type AIZO and $Al_2O_3$/IZO TFTs were fabricated using a heavily doped p-type Si substrate covered with thermally oxidized $SiO_2$ (150 nm thick). A p-type Si substrate and a $SiO_2$ layer were used as the gate electrode and gate insulator, respectively. First, as the reference channel layer, 25 nm amorphous AIZO layers were deposited onto the substrate at a room temperature of 25 °C using $Al_2O_3$ (with a purity of 99.99%) and IZO ($In_2O_3$: ZnO = 90 wt.%: 10 wt.%) targets via radio frequency (RF) magnetron co-sputtering. The $Al_2O_3$ target power was varied from 0 to 30 W in 10 W increments to confirm the effect of the Al cations in the channel layer. The IZO target power was fixed at 50 W, and the OPP [$O_2$/(Ar + $O_2$)] in the chamber was set to 13%.

The $Al_2O_3$/IZO channel layers were then deposited with a dual-layer structure as the experimental channel layer. At room temperature, a 20 nm IZO front-channel layer was deposited onto the substrate using the IZO target with 50 W power under 13% OPP via RF magnetron sputtering. An $Al_2O_3$ back-channel layer with less than 5 nm thickness was deposited onto the front-channel layer by applying power to the $Al_2O_3$ target under 13% OPP conditions via RF magnetron sputtering, increasing the power in 10 W steps from 20 to 50 W. This was performed to evaluate the effect of the $Al_2O_3$ back-channel target power on the electrical characteristics and gate bias stability of the fabricated TFTs. In addition, to verify the effect of the OPP condition in the chamber, we deposited an $Al_2O_3$ back-channel layer using an $Al_2O_3$ target with 50 W power under 13%, 20%, 25%, and 33% OPP conditions. During channel layer deposition, the initial pressure of the chamber was $3 \times 10^{-6}$ Torr, and the process pressure was $2 \times 10^{-3}$ Torr. In addition, all channel layers were patterned at $1000 \times 2000$ μm using a shadow mask. After depositing the channel layers, annealing was performed using a hot plate at 250 °C for 1 h.

Next, 100 nm-thick Al was deposited using a shadow mask with a thermal evaporator to serve as the source/drain (S/D) electrodes. The fabricated TFTs had channel widths (W) and lengths (L) of 1000 and 100 μm, respectively. Figure 1 shows a schematic cross-section of the fabricated inverted-staggered-type (a) AIZO and (b) $Al_2O_3$/IZO TFTs. The electrical characteristics and bias stability of the fabricated TFTs were measured using an EL423 (ELESC) semiconductor parameter analyzer with two probes.

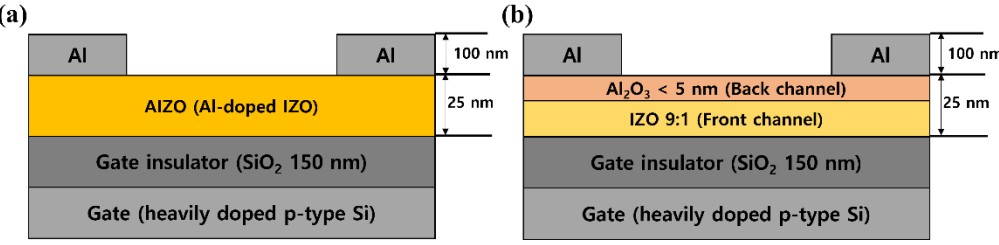

**Figure 1.** Schematic cross-section of inverted-staggered-type (**a**) AIZO and (**b**) $Al_2O_3$/IZO TFTs.

In addition, X-ray photoelectron spectroscopy (XPS) depth profiling analysis was performed to confirm the distribution of Al cations according to the depths in the AIZO and

$Al_2O_3$/IZO channel layers. Each XPS analysis was performed while dry etching the channel layers at intervals of 5 nm from the surface. The XPS measurements were performed using a monochromated Al K$\alpha$ X-ray source (hv = 1486.6 eV) at 15 kV/150 W. The spot size used was 400 μm (Theta Probe AR-XPS System, Thermo Fisher Scientific, Waltham, MA, USA). The O 1 s spectra were deconvoluted into two peaks, centered at 530.1 $\pm$ 0.1 (OL) and 531.7 $\pm$ 0.1 eV (OH) [19,20].

## 3. Results and Discussion

Figure 2a and Table 1 show the transfer curves and electrical characteristic parameters of the co-sputtered AIZO TFTs, respectively, with respect to the $Al_2O_3$ target power. The AIZO 0 W (=pristine IZO) TFTs, which did not contain Al cations in the channel layer, exhibited poor electrical characteristics, except for high electron mobility. By increasing the $Al_2O_3$ target power during co-sputtering, the number of Al cations in the channel layer increased, suppressing the oxygen vacancy generation. Therefore, the electron mobility and off-current decreased slightly, and the threshold voltage shifted positively owing to the decreased carrier concentration and traps. Compared to the IZO TFTs, the AIZO TFTs exhibited improved bias stability. However, the optimized AIZO TFT with an $Al_2O_3$ target power of 20 W exhibited poor switching characteristics (=subthreshold swing) of 1.30 V/decade. The Al cations in the channel layer easily migrated to the channel–insulator interface and served as traps. However, in the $Al_2O_3$/IZO TFTs, the Al cations diffused into the IZO front-channel layer during annealing, which served as the buffer layer. The number of Al cations at the channel–insulator interface was significantly lower than that of AIZO TFTs.

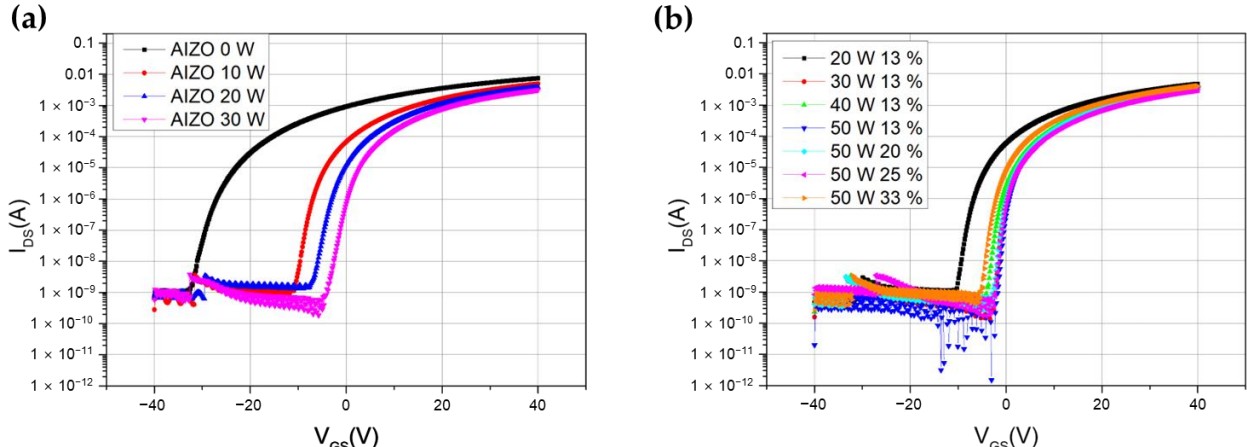

**Figure 2.** Transfer curves of (**a**) co-sputtered AIZO TFTs and (**b**) $Al_2O_3$/IZO TFTs according to $Al_2O_3$ target power and OPP in chamber.

**Table 1.** Electrical parameters of co-sputtered AIZO TFTs according to $Al_2O_3$ target power.

| $Al_2O_3$ Power (W) | $\mu_{sat}$ (cm$^2$/V·s) | $V_{th}$ (V) | $I_{on}/I_{off}$ Ratio | SS (V/Dec) |
|---|---|---|---|---|
| **0 (Pristine IZO)** | 41.48 | −26.88 | $9.8 \times 10^6$ | 1.44 |
| **10** | 27.25 | −5.12 | $6.3 \times 10^6$ | 1.14 |
| **20** | 23.61 | −0.96 | $2.9 \times 10^6$ | 1.30 |
| **30** | 22.29 | 0.96 | $1.1 \times 10^7$ | 1.29 |

Figure 2b and Table 2 show the transfer curves and electrical properties of the fabricated $Al_2O_3$/IZO TFTs according to the $Al_2O_3$ target power and OPP in the chamber, respectively. During $Al_2O_3$ back-channel deposition, the deposition rate increased in proportion to the $Al_2O_3$ target power. The $Al_2O_3$/IZO TFTs, fabricated using 20 W $Al_2O_3$ target power under the 13% OPP condition, exhibited worse characteristics than those under other conditions. This difference was attributed to the insufficient total amount of Al

cations diffused from the back channel to the front channel during annealing. However, this device exhibited relatively high electron mobility. This is because, as an insufficient amount of Al cations diffused into the IZO channel layer, the s-orbital overlap of indium and zinc cations was less disturbed compared to the other conditions [11–13]. The amount of Al cations diffused into the IZO front channel was increased by raising the $Al_2O_3$ target power while fixing the OPP to 13% to improve the electrical characteristics of the fabricated TFTs. As the target power increased, the on–off current ratio ($I_{on}/I_{off}$) and subthreshold swing (SS) improved significantly. In addition, it was confirmed that the saturation mobility ($\mu_{sat}$) was maintained above 20 cm$^2$/V·s, and the threshold voltage ($V_{th}$) positively shifted to near 0 V.

**Table 2.** Electrical parameters of $Al_2O_3$/IZO TFTs according to $Al_2O_3$ target power and OPP in chamber.

| $Al_2O_3$ Power (W) | OPP (%) | $\mu_{sat}$ (cm$^2$/V·s) | $V_{th}$ (V) | $I_{on}/I_{off}$ Ratio | SS (V/Dec) |
|---|---|---|---|---|---|
| 20 | 13 | 26.44 | −5.76 | $6.6 \times 10^6$ | 1.01 |
| 30 | 13 | 23.04 | 0.96 | $7.3 \times 10^6$ | 1.00 |
| 40 | 13 | 21.16 | 0.16 | $7.9 \times 10^6$ | 0.80 |
| 50 | 13 | 23.34 | 0.96 | $6.8 \times 10^7$ | 0.61 |
| 50 | 20 | 21.94 | 1.12 | $1.2 \times 10^7$ | 0.70 |
| 50 | 25 | 19.81 | 1.6 | $1.2 \times 10^7$ | 0.69 |
| 50 | 33 | 24.49 | −0.8 | $6.3 \times 10^6$ | 0.86 |

Because the co-sputtered AIZO channel layer, which was deposited using an IZO (In: Zn = 90 wt.%: 10 wt.%) target, exhibited optimal performance when the OPP in the chamber was 13%, there existed limitations to improving the electrical properties by changing the OPP conditions. However, in the $Al_2O_3$/IZO channel structure, as the IZO front-channel layer was first deposited under 13% OPP, the $Al_2O_3$ back-channel layer was deposited by varying the OPP in the chamber. To verify the effect of the OPP in the chamber during $Al_2O_3$ back-channel deposition, we analyzed the electrical properties of the fabricated $Al_2O_3$/IZO TFTs by fixing the target power to 50 W and increasing the OPP in the chamber (Figure 2b and Table 2). As the OPP increased, the carrier mobility decreased slightly, the threshold voltage shifted in the positive direction, the on–off current ratio was approximately $10^7$, and the SS values ranged between 0.6 and 0.7 V/dec. Without this tendency, the electrical characteristics of the fabricated TFTs deteriorated when the OPP was 33%. When the OPP was increased under an identical process pressure, the number of Ar atoms in the chamber and the sputtering rate decreased; therefore, the $Al_2O_3$ back-channel layer contained an insufficient amount of Al cations.

Finally, we fabricated improved $Al_2O_3$/IZO TFTs with 50 W $Al_2O_3$ target power and 13% OPP in the chamber during back-channel deposition. In Figure 3, we denote the pristine IZO TFTs, Al-doped IZO TFTs with 20 W $Al_2O_3$ target power, and $Al_2O_3$/IZO TFTs with 50 W $Al_2O_3$ target power under the 13% OPP condition as "AIZO 0 W", "AIZO 20 W", and "$Al_2O_3$/IZO Dual Layer", respectively. The results show that the $Al_2O_3$/IZO TFTs exhibited a high electron carrier mobility of 23.34 cm$^2$/V·s and a low threshold voltage of 0.96 V. In particular, compared to AIZO 20 W TFTs, the SS value representing the switching characteristics of TFTs decreased by more than 53% to 0.61 V/dec, and the on–off current ratio increased by more than 20 times to $6.8 \times 10^7$. Therefore, in the fabricated $Al_2O_3$/IZO TFTs, compared to the co-sputtered AIZO TFTs, the electrical properties were adjusted by varying the $Al_2O_3$ target power and the OPP during back-channel deposition. The switching characteristics, which are the disadvantages of conventional AIZO TFTs, were improved, and a high on–off current ratio was obtained.

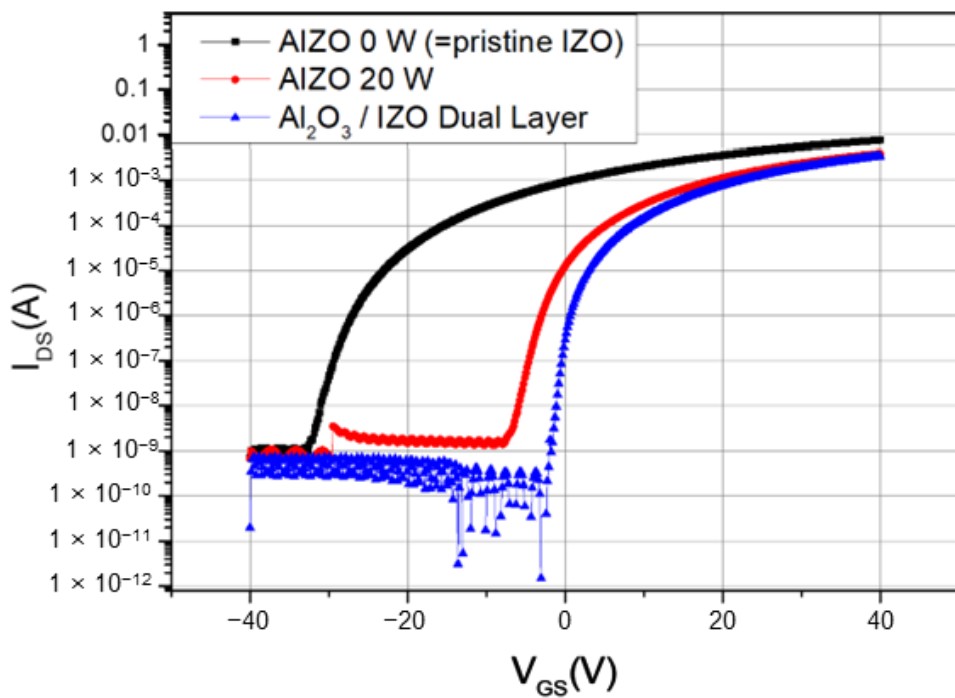

**Figure 3.** Transfer curves of AIZO 0 W ($Al_2O_3$ target power = 0 W, OPP = 13%), AIZO 20 W ($Al_2O_3$ target power = 20 W, OPP = 13%), and $Al_2O_3$/IZO Dual Layer ($Al_2O_3$ target power = 50 W, OPP = 13%) TFTs.

However, the gate bias stability of the $Al_2O_3$/IZO TFTs fabricated with 50 W $Al_2O_3$ target power and 13% OPP was degraded compared to that of the AIZO 20 W TFTs. Positive bias stress ($V_{GS}$ = +10 V, $V_{DS}$ = 0 V, PBS) and negative bias stress ($V_{GS}$ = −10 V, $V_{DS}$ = 0 V, NBS) were applied to the AIZO 20 W and $Al_2O_3$/IZO TFTs at room temperature under ambient conditions for 3600 s, respectively, to investigate the gate bias stability of the fabricated TFTs. During the PBS test, the AIZO 20 W and $Al_2O_3$/IZO TFTs showed a threshold voltage shift of approximately +1 V, exhibiting relatively good gate bias stability. However, during the NBS test, the AIZO 20 W and $Al_2O_3$/IZO TFTs showed threshold voltage shifts of −3.36 and −4.96 V, respectively, indicating poor gate bias stability. The PBS test results were better than the NBS test results because the AIZO 20 W and $Al_2O_3$/IZO channel layers had more traps in the back channel than in the channel–insulator interface [21,22]. To improve the gate bias stability of the $Al_2O_3$/IZO TFTs under NBS, we reduced the oxygen vacancies in the back channel and analyzed the threshold voltage shift by varying the $Al_2O_3$ target power and OPP in the chamber during back-channel deposition in Figure 4. Under 13% OPP, all the NBS results were poor because of numerous oxygen vacancies in the back-channel layer (Figure 4a). In contrast, under 20% OPP, the NBS results improved as the $Al_2O_3$ target power increased, and the best results were obtained at 50 W target power. The NBS results improved as the target power increased because the Al cations in the back-channel increased, and the traps on the surface decreased. Because the applied power limit of the two-inch $Al_2O_3$ target was 60 W, the target power was applied from 0 to 50 W in 10 W increments.

To further improve the gate bias stability under NBS, we analyzed the threshold voltage shift according to the OPP in the chamber during back-channel deposition while fixing the $Al_2O_3$ target power to 50 W (Figure 4b). Below 20% OPP, the NBS results improved as the OPP increased, and the NBS threshold voltage shift under 20% OPP showed the lowest value of −2.08 V. As the OPP increased during back-channel deposition, the number of oxygen vacancies on the surface decreased. Above 20% OPP, the NBS results deteriorated as the OPP increased up to 33%. As explained earlier (Figure 2), the number of Ar atoms in the chamber and the sputtering rate of the $Al_2O_3$ target decreased with increasing OPP under an identical process pressure. Therefore, the Al cations in the

back-channel were insufficient in suppressing oxygen vacancy generation. Based on the electrical performance and gate bias stability, optimized $Al_2O_3/IZO$ TFTs were fabricated with an $Al_2O_3$ target power of 50 W and 20% OPP during back-channel deposition. The PBS and NBS results of the AIZO 20 W and optimized $Al_2O_3/IZO$ TFTs are presented in Figure 5 and Table 3. The optimized $Al_2O_3/IZO$ TFTs exhibited better gate bias stability than the AIZO 20 W TFTs under PBS and NBS. The PBS and NBS results improved from +1.44 to +0.32 V and −3.36 to −2.08 V, respectively.

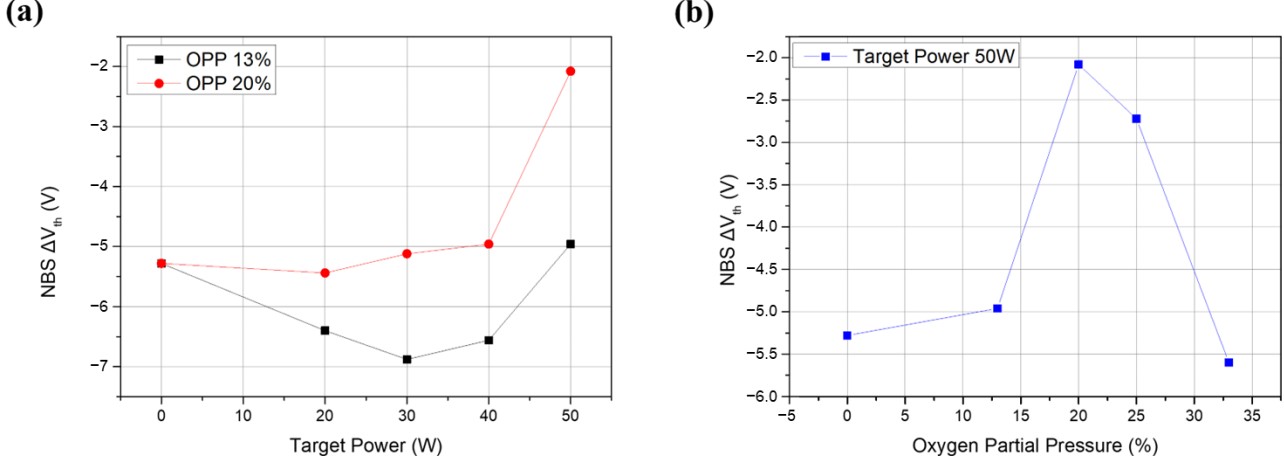

**Figure 4.** Comparison of NBS results according to (**a**) $Al_2O_3$ target power and (**b**) OPP.

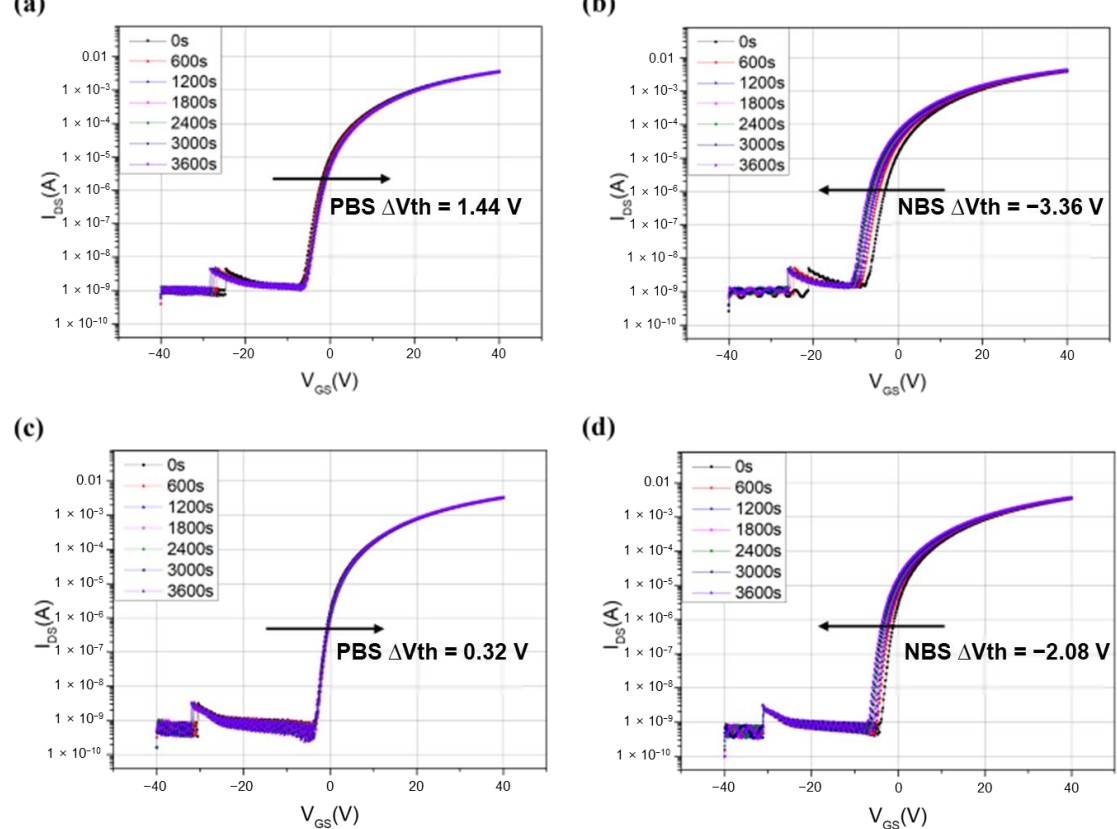

**Figure 5.** Gate bias stress results for AIZO and $Al_2O_3/IZO$ TFTs for varying stress durations: (**a**) PBS: AIZO 20 W, (**b**) NBS: AIZO 20 W, (**c**) PBS: $Al_2O_3/IZO$ 50 W OPP 20%, and (**d**) NBS: $Al_2O_3/IZO$ 50 W OPP 20%.

**Table 3.** $V_{th}$ shifts in AIZO and $Al_2O_3$/IZO TFTs under PBS and NBS conditions (over 3600 s).

| Type of Stress | AIZO TFT ($Al_2O_3$ = 20 W) | $Al_2O_3$/IZO TFT ($Al_2O_3$ = 50 W, OPP = 20%) |
|---|---|---|
| $\Delta V_{th}$ of PBS (V) | +1.44 | +0.32 |
| $\Delta V_{th}$ of NBS (V) | −3.36 | −2.08 |

To investigate the effect of the $Al_2O_3$/IZO channel structure on electrical performance and gate bias stability, we performed an XPS depth profiling analysis to investigate the chemical bonding states according to the depths in the active-channel layers of the AIZO 20 W and optimized $Al_2O_3$/IZO TFTs ($Al_2O_3$ target power: 50 W, OPP: 20%). Al cations, the carrier suppressors, exist in about 1 wt.% in the channel layers. Their chemical composition is very small, so it is difficult to directly measure them [13]. Therefore, in the XPS depth profiling results, we tried to indirectly confirm the distribution of Al cations, the carrier suppressors, by showing changes in oxygen binding states according to the depths of the AIZO and $Al_2O_3$/IZO channel layers. The results are presented in Figure 6 and Table 4. The lower binding energy peak area ($O_L$) in red and the higher binding energy peak area ($O_H$) in blue are related to the oxygen anions present in the metal–oxide bonds (M-O bonds) and the metal oxide lattice with oxygen vacancies ($V_O$), respectively [19,20].

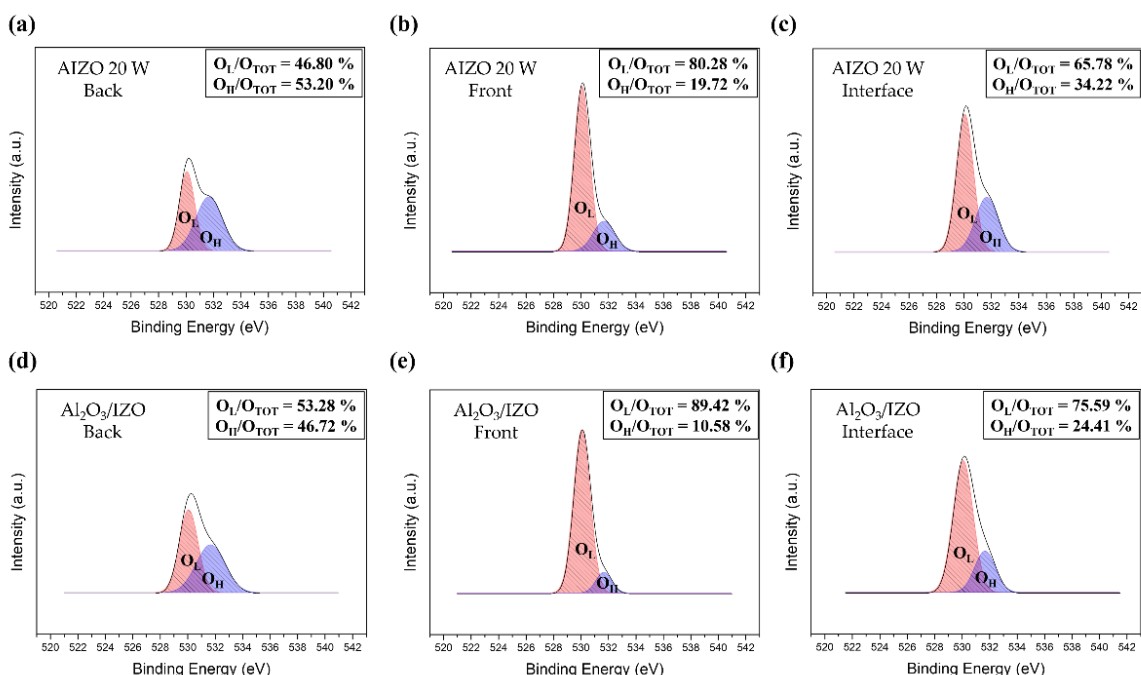

**Figure 6.** XPS of O 1s region in (**a**) AIZO 20 W back channel, (**b**) AIZO 20 W front channel, (**c**) AIZO 20 W channel–insulator interface, (**d**) $Al_2O_3$/IZO 50 W OPP 20% back channel, (**e**) $Al_2O_3$/IZO 50 W OPP 20% front channel, and (**f**) $Al_2O_3$/IZO 50 W OPP 20% channel–insulator interface.

**Table 4.** XPS analysis results for AIZO and $Al_2O_3$/IZO TFTs according to channel layer depth.

| Depth of channel | AIZO TFT ($Al_2O_3$ = 20 W) | | $Al_2O_3$/IZO TFT ($Al_2O_3$ = 50 W, OPP = 20%) | |
|---|---|---|---|---|
| | $O_H/O_{TOT}$ | $O_L/O_{TOT}$ | $O_H/O_{TOT}$ | $O_L/O_{TOT}$ |
| **0 nm (Back channel)** | 53.20% | 46.80% | 46.72% | 53.28% |
| **10 nm (Front channel)** | 19.72% | 80.28% | 10.58% | 89.42% |
| **20 nm (Channel–insulator interface)** | 34.22% | 65.78% | 24.41% | 75.59% |

The back channel, front channel, and channel–insulator interface indicated that the active-channel layer was located at depths of 0, 10, and 20 nm from the surface, respectively. In the front channel and channel–insulator interface, the $Al_2O_3$/IZO TFTs showed a lower ratio of the OH peak area to the total-binding energy peak area (OTOT) than the AIZO 20 W TFTs, as shown in Figure 6b,c,e,f. These results demonstrate that the $Al_2O_3$/IZO channel layer has chemical bonds with fewer oxygen vacancies than the co-sputtered AIZO channel layer. Therefore, the fabricated $Al_2O_3$/IZO TFTs exhibited improved switching characteristics, a high on–off current ratio, and good gate bias stability under PBS. In addition, for the back channel results, the co-sputtered AIZO channel layer showed a higher OH/OTOT peak area ratio than the OL/OTOT peak area ratio. This suggests that the number of M-O bonds is fewer than that of the oxygen vacancies on the back-channel surface, as shown in Figure 6a. However, the $Al_2O_3$/IZO had a lower OH/OTOT peak area ratio than the OL/OTOT peak area ratio (Figure 6d). These results indicate that the $Al_2O_3$/IZO channel layer, in which the back channel was deposited at 50 W $Al_2O_3$ target power under 20% OPP, had fewer oxygen vacancies on the back-channel surface than the co-sputtered AIZO channel layer, exhibiting an improved gate bias stability under NBS. From the XPS results, we confirmed that the $Al_2O_3$/IZO channel structure suppressed the oxygen vacancies in the entire range of the channel layer compared to the AIZO channel layer, improving the electrical performance and gate bias stability.

## 4. Conclusions

We fabricated AOS TFTs with an $Al_2O_3$/IZO channel structure to improve the switching characteristics and gate bias stability of the co-sputtered AIZO TFTs while maintaining excellent electrical performance. In the co-sputtered AIZO channel layer, the Al cations acted as traps at the channel–insulator interface and deteriorated the switching characteristics. In the $Al_2O_3$/IZO channel layer, Al cations diffused into the front channel from the back channel during annealing by separately depositing the IZO front channel and the $Al_2O_3$ back channel such that relatively few Al cations existed at the channel–insulator interface. In addition, the electrical characteristics and gate bias stability of the $Al_2O_3$/IZO TFTs were improved by varying the target power and OPP conditions in the chamber during $Al_2O_3$ back-channel deposition. Based on the electrical characteristics only, the $Al_2O_3$/IZO TFTs, which were fabricated under 50 W $Al_2O_3$ target power and 13% OPP conditions, exhibited a high electron mobility ($\mu_{sat}$) of 23.34 $cm^2$/V·s and a low threshold voltage ($V_{th}$) of 0.96 V. Compared to the co-sputtered AIZO TFTs, the subthreshold swing, which indicates the switching characteristics, and the on–off current ratio improved significantly by more than 53% (0.61 V/dec) and 20 times ($6.8 \times 10^7$), respectively. However, for the NBS test results, the $Al_2O_3$/IZO TFTs fabricated under the 13% OPP condition exhibited a high threshold voltage shift. Therefore, considering the gate bias stability with a slight decrease in the electrical performance, the optimized $Al_2O_3$/IZO TFTs were fabricated under 50 W $Al_2O_3$ target power and 20% OPP conditions. Compared to the co-sputtered AIZO TFTs, the threshold voltage shift of the PBS and NBS decreased from +1.44 to +0.32 V and −3.36 to −2.08 V, respectively.

**Author Contributions:** Conceptualization, experiments, and writing—original draft preparation, S.-H.L.; writing—review and editing, S.-Y.B.; supervision, M.Y. All authors have read and agreed to the published version of the manuscript.

**Funding:** This study was supported by BK21PLUS, Creative Human Resource Education and Research Programs for ICT Convergence in the 4th Industrial Revolution. This study was supported by a National Research Foundation of Korea (NRF) grant funded by the Korean government (MSIT) (No. 2021R1A4A102708711). This study was supported by Korea Institute for Advancement of Technology (KIAT) grant funded by the Korea Government (MOTIE) (G02P07820002113, The Competency Development Program for Industry Specialist).

**Institutional Review Board Statement:** Not applicable.

**Informed Consent Statement:** Not applicable.

**Data Availability Statement:** Not applicable.

**Conflicts of Interest:** The authors declare no conflict of interest.

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
