# Peer review of "Improved Performance and Bias Stability of Al2O3/IZO Thin-Film Transistors with Vertical Diffusion"

_electronics, doi:10.3390/electronics11142263_

Round 1
Reviewer 1 Report
Please find the attachment.

Reviewer 2 Report
I have read the whole manuscript in detail. I am sure that it would be a meaningful research paper in Electronics if you deal with these following suggestions carefully and make corrections.
1. In the case of the abstrat, please write in the present tense. It is highly required that the general review of basic spelling and spacing between letters/numbers.
2. The device fabrication and characterization process must be explained in very detail. Detailed explanations such as which equipment/instrument was used and which channel layers were clearly patterned through conventional lithography are essential.
3. When the Al2O3 layer is deposted thinly as a dual layer atop of IZO layer, is there any deformation or thermal damage on the surface of the firstly deposited IZO layer?
4. Authors said that a layer of 5 nm or less is deposted for DAL formation. If it is deposted way less than 5 nm and a more than 5 nm, what will be happened to the electrical performance and stability of the TFTs regarding the vertical diffusion?
5. It is highly recommended that to clarify how far Al cations have been disfussed and distributed thorughout the IZO semiconducting layer. If the degree of the vertical diffusion is too strong, it does not seem much different from the co-sputtered AIZO layer. For instance, XPS depth profiling or TEM-EDS elemental mapping could help to verify the direct observation and insight of vertically diffusional behviors of Al cations into the IZO layer. (plus, add appropriate reference citations about the Al cation diffusion into any other oxide layers)
6. Please explain and introduce in detail in section '2. Mterials and Methods' such as the XPS analysis, characterizations, calibrations, peak fitting conditions and so on. (plus, add appropriate reference citations)
7. In Figure 6, it would be better to have the simple denotations at every XPS plots. It will be convenient for readers. In addition, authors should modify the XPS plots in uniform (e.g., X and Y- scale range, baseline positions, etc.).
Reviewer 3 Report
In this study, authors studied on IZO/Al2O3 bilayer structure TFTs. Using the bilayer structure, authors achieved high performance and stability. To clarify the reason for the improvement of TFT performance and stability, authors compared IZO/Al2O3 and AIZO. However, I feel that there are serious inconsistencies between the obtained experimental data and the relevant explanation. I think the obtained result itself is good, but the manuscript must be thoroughly revised. Please see my comments below.
1. In this work, authors said Al ions diffused into IZO layer by Al2O3 sputtering. However, I think the diffusion length of Al into IZO is not sufficiently long to affect the TFT performance. Authors should show how much Al diffused into IZO layer by chemical composition depth profile data.
2. In general, TFT performance is determined mainly by several nanometers of front channels. Authors should explain more why the backchannel Al2O3 affect a lot the TFT performances.
3. I think that this study mainly discusses IZO TFT and the Al2O3 plays a role of passivation layer. In this respect, this paper must be rewritten focusing on the passivation role of Al2O3 in IZO TFT. Relevantly, the term of “dual active layer” seems not correct.
4. Authors should refer to recently reported relevant literatures; Shiah et al, Nature Electronics 4 (2021) 800–807, IEEE Electron Device Letters 42 (2021) 1319-1322.
5. The most serious point is the part relevant to AR-XPS. In this study, authors employed AR-XPS measurement to investigate how physical properties are differ between front- and back-channels. However, for general AR-XPS, the penetration depth is only several nanometers. Noted that the total thickness exceeds 25 nm. Therefore, the AR-XPS data still reflects the surface region. I cannot understand why authors could conclude that the obtained results are attributed to back-channel, front-channel, and insulator interface, respectively.
Round 2
Reviewer 1 Report
Please put the value of drain-source voltage (Vds) for all transfer curves (Id-Vg characteristicts). Now the manuscript looks good.
Reviewer 2 Report
It is still highly required that the general review of the spelling and expressions in the manuscript. During the proof, the authors should revise and correct the wrong English expression such as 'OTOT? or OL?' (superscripts, subscripts, etc.,).
Reviewer 3 Report
I carefully checked the response letter and the revised manuscript. Authors well responded to all my concerns. I suggest the publication of this paper with no further revision.